Potential problems of removing one invasive species at a time: a meta-analysis of the interactions between invasive vertebrates and unexpected effects of removal programs

Ballari Sebastián A. sebastianballari@gmail.com 1
Kuebbing Sara E. 2
Nuñez Martin A. 3
1 Parque Nacional Nahuel Huapi (CENAC-APN), Consejo Nacional de Investigaciones Científicas y Técnicas (CONICET) , San Carlos de Bariloche , Río Negro , Argentina
2 School of Forestry & Environmental Studies, Yale University , New Haven , CT , United States
3 Grupo de Ecología de Invasiones, INIBIOMA, Consejo Nacional de Investigaciones Científicas y Técnicas (CONICET) , San Carlos de Bariloche , Río Negro , Argentina
Sánchez Marta
Electronic publication date: 2016 Jun 2
Publication date: 2016
Volume: 4
Electronic Location ID: e2029
Received 2016 Jan 15; Accepted 2016 Apr 19
Copyright: ©2016 Ballari et al.
Copyright year: 2016
Copyright holder: Ballari et al.
License: This is an open access article distributed under the terms of the Creative Commons Attribution License, which permits unrestricted use, distribution, reproduction and adaptation in any medium and for any purpose provided that it is properly attributed. For attribution, the original author(s), title, publication source (PeerJ) and either DOI or URL of the article must be cited.
License URL: https://creativecommons.org/licenses/by/4.0/

Keywords: Animals, Co-occurrence, Carnivores, Invasional meltdown, Nonnative, Meta-analysis, Conservation, Wildlife management

Funding: CONICET SA Ballari and MA Nuñez received support from CONICET. The funders had no role in study design, data collection and analysis, decision to publish, or preparation of the manuscript.

==============================
Although the co-occurrence of nonnative vertebrates is a ubiquitous global phenomenon, the study of interactions between invaders is poorly represented in the literature. Limited understanding of the interactions between co-occurring vertebrates can be problematic for predicting how the removal of only one invasive—a common management scenario—will affect native communities. We suggest a trophic food web framework for predicting the effects of single-species management on native biodiversity. We used a literature search and meta-analysis to assess current understanding of how the removal of one invasive vertebrate affects native biodiversity relative to when two invasives are present. The majority of studies focused on the removal of carnivores, mainly within aquatic systems, which highlights a critical knowledge gap in our understanding of co-occurring invasive vertebrates. We found that removal of one invasive vertebrate caused a significant negative effect on native species compared to when two invasive vertebrates were present. These unexpected results could arise because of the positioning and hierarchy of the co-occurring invasives in the food web (e.g., carnivore–carnivore or carnivore–herbivore). We consider that there are important knowledge gaps to determinate the effects of multiple co-existing invaders on native ecosystems, and this information could be precious for management.

Introduction

Invasive vertebrates can alter native communities and ecosystems through many pathways including predation, competition, reducing food web complexity, hybridization, competitive exclusion, and increasing the risk of extinction of native species (White et al., 2008; Doherty et al., 2015; Houde, Wilson & Neff, 2015). Many ecosystems now host numerous invasive species that directly or indirectly interact with one another and impact native species populations and ecosystem processes (Courchamp et al., 2011; Porter-Whitaker et al., 2012; Meza-Lopez & Siemann, 2015). Interactions between these co-occurring invaders are of superlative interest for wildlife management because managers can often only control or eradicate a single invasive species at a time (Glen et al., 2013). Without prior knowledge of invader interactions, removal of only a single invader can lead to an increase in the population size of other invasives or a decrease in the population size of native species (Zavaleta, Hobbs & Mooney, 2001; Campbell et al., 2011; Ruscoe et al., 2011).

Predicting the community-level consequences of management of a single invasive species requires an understanding of both the interactions between co-occurring invaders and their combined impacts (Van Zwol, Neff & Wilson, 2012; Latorre, Larrinaga & Santamaría, 2013). In an initial review of 45 invasive animal interaction studies, Jackson (2015) showed that the combined ecological impacts of multiple invaders were additive, but the mean effect size was non-additive and lower than predicted. This analysis included many animal taxonomic groups (with no mammalian cases) and ∼96% the reported interactions were from aquatic environments. In our study, we focus on invasive vertebrates because it is a homogeneous group to compare and includes some of the most damaging and widespread invasive species that are frequent targets for management (White et al., 2008; Dawson et al., 2015).

Interactions between nonnative species can be positive, negative, or neutral (Kuebbing & Nuñez, 2015; Jackson, 2015; Doherty et al., 2015). Most research on invasive species interactions has focused on facilitative interactions (i.e., invasional meltdown hypothesis, Simberloff & Von Holle, 1999; Simberloff, 2006), the replacement of one invasive by another invasive (Lohrer & Whitlatch, 2002), or mechanics that involve negative interactions, such predation (e.g., hyperpredation) and competence (e.g., mesopredator release) (Blanco-Aguiar et al., 2012; Doherty et al., 2015; Ringler, Russell & Le Corre, 2015).

Figure 1 Hypothetical food interaction webs with co-occurring native and invasive species adapted from Zavaleta, Hobbs & Mooney (2001).

The trophic level of co-occurring invaders could influence outcomes when a single invasive species is removed (red cross; B–D). In (A) hypothetical food web based in interaction of carnivore top predators, omnivores, herbivore preys and plants. In (B) the removal of a carnivore releases nonnative herbivores, and native omnivores and predators. In (C) the removal of a nonnative herbivore reduces population size of the competing native herbivore. In (D) the removal of only one invasive carnivore releases the other invasive carnivore predating on native herbivores and native omnivores reducing their populations. Thicker lines represent larger population sizes.

Many ecosystems host numerous species with different trophic positions that make up a complex network interactions (Fig. 1A; Zavaleta, Hobbs & Mooney, 2001). It may be possible to predict these type of interactions between vertebrate invaders and their potential impacts because the interactions among multiple invasive species should vary depending on the traits, trophic positions, and interactions of the co-occurring invasive and native species in the community (Fig. 1A; Zavaleta, Hobbs & Mooney, 2001; Roemer, Donlan & Courchamp, 2002; Didham et al., 2009; Oyugi, Cucherousset & Britton, 2012). For example, two invasive carnivores occupying the same trophic position may predate on similar native species or utilize similar habitats, which could lead to both invaders investing energy to compete against one another (Fig. 1B; Griffen, Guy & Buck, 2008). Thus, the removal of only one invasive predator could release the population of the second invasive predator (i.e., mesopredator release), which could ultimately cause a greater impact on the native prey species (Courchamp, Langlais & Sugihara, 1999). We may also expect different outcomes of single-species management when multiple co-existing invasive species occupy different positions in food webs (Figs. 1C and 1D; Zavaleta, Hobbs & Mooney, 2001). In a hypothetical coexistence scenario of an invasive carnivore predator and an invasive herbivore, we might expect that the removal of the invasive carnivore could reduce predation pressure on the invasive herbivore prey and allow its population to increase (Fig. 1C; Bergstrom et al., 2009). The consequence of this herbivore release may indirectly affect native herbivores through competition, or directly threaten a native plant through herbivory (Fig. 1C, Vázquez, 2002). On the other hand, if the removed species is an invasive herbivore prey, the invasive carnivore predator would be forced to change their diet and search for native prey (i.e., hyperpredation, Fig. 1D; Bate & Hilker, 2012). These hypothetical examples illustrate how the coexistence of invasive vertebrates and subsequent removal of one of them can lead to predictable impacts on native biodiversity (Zavaleta, Hobbs & Mooney, 2001).

We assessed whether the trophic positions of invasive vertebrates could predict the consequences of removal of only a single invasive species on native species. To do this, we conducted an extensive literature search of studies that evaluated the impact of removing a single invasive vertebrate while leaving a second invasive present on native biodiversity. We focused on invasive vertebrates owing to their biological and socioeconomic importance and because there are still many gaps of information on management of invasive vertebrates. We ask (1) what is the combined effect of two invasive vertebrate species on native biodiversity relative to a single invasive vertebrate? (2) does the removal of a single invasive vertebrate reduce the impact on native species? and finally (3) what traits of invasive vertebrate species (e.g., trophic position) predict these interactions?

Figure 2 Flow diagram.

A flow diagram of the screening protocol for paper selection in this study (from Moher D, Liberati A, Tetzlaff J, Altman DG, The PRISMA Group (2009). Preferred reporting items for systematic reviews and meta-analyses: the PRISMA statement. PLoS Med 6(6): e1000097. doi:10.1371/journal.pmed1000097).

Table 1 List of references used in this study for meta-analysis.

No	Reference	Title	Journal	Location	
1	Didham et al. (2009)	The interactive effects of livestock exclusion and mammalian pest control on the restoration of invertebrate communities in small forest remnants	New Zealand Journal of Zoology	Waikato region, New Zealand	
2	Houde, Wilson & Neff (2015)	Competitive interactions among multiple invasive salmonids and two populations of Atlantic salmon	Ecology of Freshwater Fish	Ontario, Canada	
3	Latorre, Larrinaga & Santamaría (2013)	Combined impact of multiple exotic herbivores on different life stages of an endangered plant endemism, Medicago citrina	Journal of Ecology	Cabrera Island, Spain	
4	Oyugi, Cucherousset & Britton (2012)	Temperature-dependent feeding interactions between two invasive fishes competing through interference and exploitation	Reviews in Fish Biology and Fisheries	United Kingdom	
5	Porter-Whitaker et al. (2012)	Multiple predator effects and native prey responses to two invasive Everglades cichlids	Ecology of Freshwater Fish	Everglades, USA	
6	Smith (2005)	Effects of invasive tadpoles on native tadpoles in Florida: evidence of competition	Biological Conservation	Florida, USA	
7	Van Zwol, Neff & Wilson (2012)	The effect of invasive salmonids on social dominance and growth of juvenile atlantic salmon	Transactions of the American Fisheries Society	Ontario, Canada	
8	Wilson et al. (2006)	An experimental study of the impacts of understorey forest vegetation and herbivory by red deer and rodents on seedling establishment and species composition in Waitutu Forest, New Zealand	New Zealand Journal of Ecology	Fiordland National Park, New Zealand	

Materials & Methods

We searched for peer-reviewed literature on invasive vertebrate interactions (Fig. 2) using the database Web of Science® and the methodology proposed by Kuebbing & Nuñez (2015). We used the keywords “species” AND “invas*” OR “alien” OR “nonnative” OR “non-indigenous,” and also used as search terms the genres of mammals, birds, reptiles, amphibians and fish described in the list of 100 most damaging invasive species in Global Invasive Species Database (Global Invasive Species Database, 2015; http://www.issg.org/database/species/) and categories filter (Supplemental Information 1). From the articles returned by this search (n = 403, Fig. 2), we selected those that met the following criteria: (1) studied the impact of an invasive vertebrate on a native species; (2) included a treatment where two invasive vertebrate species were present; and (3) included a treatment where one invasive vertebrate species was removed. This selection restricted our meta-analysis to eight published studies that comprised 128 individual observations (Table 1). Finally, to investigate if there were any species or habitat characteristics that affected the type of interaction we collected the following factors for each observation: (1) trophic position (e.g., carnivore, herbivore, omnivore) of each native and invasive species; and (2) if the invasive species overlapped in their native ranges (Tables S1 and S2). We recorded the following descriptive variables: (1) invasive species studied; (2) native species studied; (3) location of study; (4) habitat type (forest, wetland, freshwater, garrigue). We estimated mean effect sizes using Hedges’ d +, which measures the difference between treatment groups (i.e., performance of a native species in the presence of one invasive species, see Table S2) and control groups (i.e., performance of a native species in the presence of two invasive species, see Table S2). This method corrects for small sample size bias and avoids overestimating effect sizes when study sample size is low (Gurevitch & Hedges, 2001; Lajeunesse & Forbes, 2003). When necessary, we extracted data with extraction software (ImageJ, version 1.449p; Wayne Rasband, Research Services Branch, National Institute of Mental Health, Bethesda, Maryland, USA). We considered all response variables in each study (e.g., if a study measured fitness and growth of a native animal). We consider a mean effect size to be significant when its 95% confidence intervals do not overlap zero. Because of potential publication bias against studies with negative results or studies with higher sample sizes having a probability of finding effects, we assessed potential publication bias by plotting the sample size against the Hedges’ d value (e.g., funnel plot analysis; Palmer, 1999). We found a funnel-shape distribution of data that is expected in the absence of publication bias (Fig. S1). Because all eight studies reported multiple response variables for the affected native species, there is a potential issue with independence among observations within a study. To avoid this problem, we also ran the meta-analysis on a reduced dataset randomly selecting a single response variable to describe the effect of the removal of a specific nonnative species on a specific native species. The mean effect sizes for the reduced dataset was similar to the mean effect size for the entire dataset, and the 95% confidence intervals overlapped for both datasets (Table 2 and Table S3). Therefore, we felt confident in including all 128 observations in our analysis.

Table 2 Results from a meta-analysis of 8 published manuscripts entailing 128 observations of invasive vertebrate interactions.

We report the mean effect size and 95% confidence intervals (Hedge’s d +) andbold values when the 95% CI does not overlap zero. Mean effect sizes were calculated for the entire data set and subsets of the data that compared the effect of mixed and single groups of invasive vertebrateson native biodiversity.

	N	Direction	Hedge’s d+	
Habitat type				
Forest	16	–	−0.29 ± 0.10	
Wetland	36	–	−0.13 ± 0.05	
Freshwater	73	–	−0.11 ± 0.05	
Garrigue	3	–	−0.16 ± 0.15	
Native range overlap				
Overlapping ranges	46	–	−0.21 ± 0.07	
Non-overlapping ranges	72	–	−0.13 ± 0.03	
Invasive functional group				
Amphibian	16	0	−0.13 ± 0.13	
Mammal	19	–	−0.25 ± 0.08	
Fish	93	–	−0.13 ± 0.03	
Trophic position of removed invader				
Carnivore	106	–	−0.13 ± 0.03	
Herbivore	6	0	−0.06 ± 0.15	
Omnivore	4	–	−0.32 ± 0.10	

Results

We found that the removal of a single invasive species always led to a negative or neutral mean effect on native species performance or survival (Fig. 3; Table 2). Surprisingly, we never found a positive effect size where the removal of one invasive led to an increase in native performance (Table 2). Related to trophic position, we found that the majority of the invasive vertebrates studied were strict carnivores (52.9%, n = 9), while the minority were herbivores (23.5%, n = 4) or omnivores (23.5%, n = 4; Table S1). Likewise, the vast majority of observations included interactions between two carnivorous species (82.8%, n = 106), while only 11 observations included interactions between an invasive herbivore and omnivore (8.6%) and a single observation between two omnivores. Of the 17 species reviewed, there were 8 fish, 6 mammals, 2 amphibians and 1 marsupial (Table S1). Regarding the location, the majority of the observations were from North America (Canada and United States, 82.8%, n = 106), while only 12.5% were in Oceania (New Zealand, n = 16) and 4.7% in Europe (United Kingdom and Spain, n = 6). Only 14.8% (n = 19) of the observations were on islands. Finally we found significantly negative mean effect sizes regardless of the whether the nonnative species overlapped in their native range, and across habitat types (Table 2).

Figure 3 Mean effect on native diversity performance or survival across all trophic levels of nonnative vertebrates.

In ecosystems invaded by two nonnative vertebrates, the removal of only a single invader had a negative mean effect on native diversity performance or survival (Hedges’ d +) across all trophic levels. Error bars represent 95% confidence intervals of the mean.

Discussion

Our results show that the removal of a single invasive species led to a negative or neutral mean effect on native species performance or survival. This could suggest, in accordance with Jackson (2015), that the interactions between vertebrate invaders are antagonistic and reduce the population size and impact of other invaders. The studies we reviewed overwhelmingly considered the effects of two carnivorous species on native prey species (82.8%, n = 106), so we may need to limit this interpretation to this particular scenario. It is likely that in scenarios where the co-occurring invaders are not competing predators (e.g., carnivore–herbivore), the positive effects on native biodiversity could occur at different trophic levels, when carnivore predator are removed (e.g., in native omnivores and plants in Figs. 1B and 1C, Zavaleta, Hobbs & Mooney, 2001; Vázquez, 2002; Griffen, Guy & Buck, 2008). In contrast, in this scenario, the removal of predator also could lead to mesopredator release (native or nonnative) to the detriment of native species (Zavaleta, Hobbs & Mooney, 2001). On other hand, when invasive herbivore is removed, plants (native -Fig. 1B- or nonnative) could have significant benefits (Courchamp, Chapuis & Pascal, 2003).

We found many gaps in our review concerning the impacts of removing a single invasive vertebrate species on native biodiversity, which highlights research areas in need of further study. The major knowledge gap is expanding our understanding of removal of herbivore and omnivore vertebrate invaders may influence other nonnative and native species in the food web. The majority of the invasive vertebrates we studied were strict carnivores and the minority were herbivores or omnivores. Likewise, most of the observations included interactions between two carnivorous species, while few recorded interactions between an invasive herbivore and omnivore or two omnivores. Globally, there are many examples of co-occurrence of invasive vertebrates that occupy these missing trophic positions (herbivorous “h”—omnivorous “o” (e.g., livestock-wild boar, Desbiez, Santos & Keuroghlian, 2009) or their combinations “h”–“h”(e.g., cattle-deer, Flueck, Franken & Smith-Flueck, 1999) 1999) or “o”–“o”(e.g., brushtail possum-black rat, Wilson et al., 2006). For example, in South America and New Zealand, large nonnative herbivores such as cattle, goat, and deer modify and alter plant communities, which affect other invasive herbivore species such as rabbits and hares, and/or omnivores like wild boar, rats, and opossums (Glen et al., 2013; Lantschner, Rusch & Hayes, 2013; Whitehead et al., 2014). However, we did not find studies that evaluated the consequences or the individual effects of single-invader eradication of these invasive species combinations. Also, the studies we found lacked information on vertebrate groups like reptiles and birds. However, in different regions of the world, several species of invasive reptiles (e.g., Python bivittatus, Varanus niloticus, Iguana iguana, in USA, Engeman et al., 2011) or invasive birds (e.g., Psittacula krameri, Acridotheres tristis, Sturnus burmannicus in Israel, Orchan et al., 2013) coexist and affect native biodiversity. Although we did not find that the removal of one invasive led to an increase in native performance, we do not think this is because this does not occur. In nature, there are many possible scenarios where the removal of an invasive species might negatively affect the presence of another invader and positively affect native biodiversity (e.g., invasive host and pathogens, invasive specialize mutualism). These gaps could contribute more insight into the implications of single-species invasive removal and potentially expand the results found in this work.

In wildlife management is crucial to understand the outcomes of the applied methods (e.g., Zavaleta, Hobbs, & Mooney, 2001), in particular the removal of only a single invasive species in a scenario with multiple invasive species (Bonnaud et al., 2010). But also, it is clear that we need more studies and experiments across different regions, invasive species combinations, interactions with different trophic positions, and management strategies to test if we can predict or anticipate the results of these invasive interactions (Smith, 2005; Bergstrom et al., 2009).

Eradications efforts are very complex owing to the fact that they need exceptional planning. Even though eradications may benefit some biological diversity, they can have unwanted and unexpected impacts on native species and ecosystems (Zavaleta, Hobbs & Mooney, 2001; Caut, Angulo & Courchamp, 2009; Ruscoe et al., 2011). We believe that when possible, management initiatives should consider integrated management of invasive species, considering trophic interactions between invaders and native species, to detect possible direct or indirect unexpected consequences for native species and ecosystems (Zavaleta, Hobbs & Mooney, 2001; Caut, Angulo & Courchamp, 2009; Ruscoe et al., 2011; Glen et al., 2013; Ringler, Russell & Le Corre, 2015).

We suggest that considering the type of interactions and trophic positions of the co-occurring invasive vertebrates might provide a predictive framework for understanding when single-species management will lead to unwanted and unexpected effects, but more data is necessary to test this hypothesis. We call for more studies of the effects of co-occurring invasive vertebrates, particularly of scenarios where invaders occupy the following trophic positions: predator–herbivore; predator–predator; predator-omnivore; omnivore-herbivore, herbivore–herbivore. These studies will clarify and bring to light possible outcomes of the removal of single-invaders on native biodiversity.

Supplemental Information

Supplemental Information 1 Literature Search

Literature search: search terms use in a Web of Science® search for peer-reviewed studies on the impacts of removal a single invasive vertebrate species on native biodiversity. Details of the search conducted on the basis of genus of the terrestrial vertebrates invasive species reported among the 100 most damaging invasive species Global Invasive Species Database.

Click here for additional data file.

Figure S1 Funnel plot analysis of sample size against the Hedges’ d value

Click here for additional data file.

Table S1 List of species and reference used for the analysis

Click here for additional data file.

Table S2 Extracted data used in meta-analysis

Click here for additional data file.

Table S3 Results from the reduced dataset* to test for issues of pseudoreplication

Click here for additional data file.

The authors thank Mariano Rodriguez Cabal for thoughtful comments on previous versions of this manuscript.

Additional Information and Declarations

Competing Interests

Author Contributions

Data Availability

The authors declare there are no competing interests.

Sebastián A. Ballari and Sara E. Kuebbing conceived and designed the experiments, performed the experiments, analyzed the data, wrote the paper, prepared figures and/or tables, reviewed drafts of the paper.

Martin A. Nuñez conceived and designed the experiments, wrote the paper, reviewed drafts of the paper.

The following information was supplied regarding data availability:

The research in this article did not generate any raw data.

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
