# Peer review of "Potential problems of removing one invasive species at a time: a meta-analysis of the interactions between invasive vertebrates and unexpected effects of removal programs"

_PeerJ, doi:10.7717/peerj.2029_

## Round 0.1 · original submission · Major Revisions

Dear Authors,

The reviewers appreciated the attention to an important topic in invasion sciences but identified some important concerns and aspects of the manuscript that should be improved. Both were bothered by the potential pseudoreplication in your data. I agree with them and consider it to be serious criticism of the manuscript. However they provide suggestions to cope with this point. Regarding the sample size (n = 8 articles), even if it is rather low, it is an acceptable number for a meta-analysis, if you have performed an appropriate systematic review of the literature before. Another important point is that authors cannot respond to the main questions of the ms whether the trophic positions of the invasive species could predict the consequences of removal of only a single invasive species on native species. As you can see, the statistical issues are substantial, so please, be aware that the inability to address these questions could lead to rejection after revision. Please, pay attention also to all the minor comments referees provided. I hope that all these detailed suggestions will be useful for improvement of the ms. On the basis of the reviewers’ report and my own review I ask for a major revision.

Kind regards

Marta I. Sánchez

Reviewer 1 ·

Basic reporting

1. Basic reporting

1.1. The article should include sufficient introduction and background to demonstrate how the work fits into the broader field of knowledge. Relevant prior literature should be appropriately referenced.
R (Reviewer): The introduction of the article has not a sufficient background and some paragraphs lack of references to prior literature. This comment refers specially to the fourth paragraphs. In my opinion the fourth paragraph should reference to Fig 1 of Zavaleta et al. 2001. The Authors Figure 2 has a high degree of similarity to Fig 1 of Zavaleta et al 2001. This paper is referenced elsewhere in the introduction, but, in my opinion, it seems to be the logic and the base of Ballari et al paper. Thus, it should be clearly state that Authors have used this figure as a base, and that they have expanded it or applied it to their specific case and question. Paragraph fourth needs to be more related to the mechanisms exposed in paragraph third (mesopredator release, invasional meltdown, or other such as hyperpredation). Strong support with references is needed (e.g. examples for each of the hypothetical cases).

1.2. Figures should be relevant to the content of the article, of sufficient resolution, and appropriately described and labeled.
R: Figures are not well labelled. There is a Figure 1 that does not correspond to the Figure 1 Authors referenced in their text. Thus, all figure numbers are incorrect.

1.3. All appropriate raw data has been made available in accordance with our Data Sharing policy.
R: I suggest Authors to include more information of the data their used. In the Supplementary file 4 (Table 3) Authors list the references used in the meta-analysis. However, the reader does not know which data was extracted and used. Specifically, I would like to know: 1) the native species and the performance traits of the native species the Authors have used, for each article. 2) Distinguish which species was the invader that was removed and which one was the invader that was not removed. 3) Clarify the trophic position of the removed invaded relatively to the other invader and relatively to the native species, as well to the trophic position of the non removed invader with respect to the native species. Thus, I suggest Authors should split the column “species” into “species removed”, “species not removed”, “native species”, and “performance trait of native species”. Besides, Authors should add their trophic position relatively to each other.

Experimental design

2. Experimental Design

2.1. The submission should clearly define the research question, which must be relevant and meaningful. The knowledge gap being investigated should be identified, and statements should be made as to how the study contributes to filling that gap.
The investigation must have been conducted rigorously and to a high technical standard.
R: I am sot sure whether the sample size used (8 publications) is enough for a meta-analysis, even when 128 individual observations were extracted. Are these 128 observations independent? Have Authors arguments to defend that their results could be generalized? I am not sure if their data is sufficient to respond to the interesting question exposed in the Introduction Section.
I am not sure if it is possible, but a way to solve this point could be to do a frequency analysis including some of the references not selected for the meta-analysis (e.g. because of the lack of response variables) but in which the Authors could maybe attempt to know whether the impact was positive, neutral or negative. Categorising the impact could increase the number of observations for other trophic position categories.

2.2. Methods should be described with sufficient information to be reproducible by another investigator.
R. Methods are not sufficiently clear:
L. 96-101. Authors should clearly state which variables will be used as factors (such those in Table 1) and which ones are descriptive variables (e.g. location of the study, name of the species). Be consistent in their names along the ms.
L.102-103. Authors should clarify which “performance” measures they have used. Related to L. 118, 135 where they state “performance or survival”. The response variable is very important and Authors should give more details and describe clearly the ones used.
L. 107-8. Authors should clarify how they treated different response variables of the same study, as correlation or non independence might occur among these data.

Validity of the findings

3. Validity of the Findings

3.1. The data should be robust, statistically sound, and controlled.
R: I doubt whether Authors can respond to the general question of the article with the data they used. As stated in their introduction L 73-4: “We assessed whether the trophic positions of invasive vertebrates could predict the consequences of removal of only a single invasive species on native species.” Thus the final goal was stated in L. 81-2: “(3) What traits of invasive vertebrate species, including trophic position, predict these interactions?” There is two comments here:

1. The 82% of the data refers to one trophic position (carnivore, L. 123-5, 138). I am not sure that the Authors could respond to their question with this heterogeneity in sample sizes among groups.

2. It is not clear whether the trophic position of the removed invader relates to the other invader, or to the native species. In Table 3 (Suppl file 4) the Author seem to refer to the trophic position of the removed invader, but it is not clear. In lines 150-154 the Authors refer to the relationships between the invasive vertebrates. In the Introduction (L.57-69) it seem equally important the trophic position that relates to the invaded non removed or to the native species.
This should be clarified and maybe it could help to reduce the heterogeneity of the sample size for the trophic position groups. It is clear that the trophic position of the invaded removed matters, but the trophic position of the non removed invader is equally important, as well as the place of the native species in relation to both invaders. Thus, Authors should include the relative trophic positions of the triangle between the three species targeted, and analyse their effects on the performance of the native species. In the Methods (L.99-100) the trophic position of the three species seem to be recorded.

3.2. The data on which the conclusions are based must be provided or made available in an acceptable discipline-specific repository.
R: Already commented in point 1.3.

3.3. The conclusions should be appropriately stated, should be connected to the original question investigated, and should be limited to those supported by the results. Speculation is welcomed, but should be identified as such.
R: Authors should ameliorate their discussion. In the present version the discussion is flaw:
The first sentences of the discussion describe briefly the results. The second paragraph lacks of references to support the statements. The third paragraph shows basically the gaps of the article. The fourth paragraph discusses results not fully supported by the results of the article and also lack of references. The last two paragraphs state the importance of the research question, which is relevant and meaningful, but which the Authors have problems to respond adequately.
L. 139-140. This sentence is difficult to understand in the context of the rest of the paragraph. Moreover, it needs some references to support it.
In general I think Authors need to discuss more their results with the previous knowledge.

3.4. Decisions are not made based on any subjective determination of impact, degree of advance, novelty, being of interest to only a niche audience, etc. Replication experiments are encouraged (provided the rationale for the replication, and how it adds value to the literature, is clearly described); however, we do not allow the ‘pointless’ repetition of well known, widely accepted results.
Negative / inconclusive results are acceptable.
R: The results of the article are inconclusive in responding the main question of the ms, but I am not sure whether the experimental design causes these results. Authors should work in identifying the categories of the main factor (trophic position) to see whether they are able to reduce the sample size heterogeneity among categories.

Additional comments

The ms entitled ‘Potential problems of removing one invasive species at a time: Interactions between invasive vertebrates and unexpected effects of removal programs” by Ballari et al. focus on detecting whether the potential impacts of the removal of an invasive vertebrate on native species could depend on the trophic position of the co-occurring invasive vertebrates. Authors use a meta-analysis to respond to this interesting question. The main result was that the removal of one invasive vertebrate causes negative effects on native species, compared to when the two invasive vertebrates are present. However, results regarding whether the trophic position matters are not conclusive, due to low sample size for some of the trophic position categories.

In the rest of my review, I follow the guidelines for reviewers of PeerJ, which is structured in three parts: 1) basic reporting, 2) experimental design and 3) validity of the findings. For each of these parts, I have commented (my text begins by “R:”) to the points requested (marked as 1.1, 1.2, etc.) only when the ms fails to meet the journal standards, and I have tried to suggest ways in which Authors could made changes to meet the appropriate standard.

Reviewer 2 ·

Basic reporting

This interesting manuscript is relatively well written overall (the language is weaker in the Discussion section), and the references are highly relevant. I have the following suggestions for improvement:
1.1. Currently, the section title “Material and Methods” is used instead of “Materials & Methods” which is the standard title for this section in PeerJ. I’d suggest to use the standard title.
1.2. The Acknowledgements section of this manuscript includes information about the study’s funding. Such information should not be provided in the Acknowledgements section in PeerJ.
1.3. Currently, the referencing of the figures is not correct, as e.g. what is meant on line 59 is not Fig. 1d but Fig. 2d –> please correct the referencing of the figures throughout the text.
1.4. Similarly, the referencing of tables needs to be corrected: in line 126, it is currently referred to Table 1, but the 17 species mentioned in this sentence are given in Table 3 in the Supplement.
1.5. The paragraph starting on line 54 and relating to Fig. 2 could be better structured, I think, particularly how it refers to Fig. 2: currently, the first mentioning of Fig. 2 is 2d. I’d normally expect to start with 2a, then 2b, c and d, rather than the other way round. It would make the flow of the text easier if this paragraph/Fig. 2 is restructured.
1.6. Comparing Figs. 2b and 2d, the only difference in the effects on remaining species seems to be that the nonnative herbivore prey increases its population size in 2b but decreases it in 2d. Please explain this difference.
1.7. Since only 8 studies were finally included in the meta-analysis, I’d suggest to cite them all in the main manuscript rather than in the Supplement. For instance, Table 2 could be moved to the main manuscript. The reason for this suggestion is that these 8 studies are obviously highly important to this meta-analysis, so I think they should be more prominently cited, i.e. cited in the main manuscript. In this way, their citations will also appear in literature databases such as the Web of Science.
1.8. Line 91: replace “the final list of articles” by “the articles returned by this search”
1.9. Line 120: replace “Related the” by “Related to”
1.10. Line 138: should be “overwhelmingly”
1.11. Lines 139/140: “likely” repeated in this sentence –> please revise
1.12. Line 145, “invasive specialize mutualism”: please correct
1.13. Line 167: should be “regardless of whether” (delete “the”)
1.14. Line 169, “if exist some relevant traits”: please correct
1.15. Line 175: “anticipate” instead of “anticipated”
1.16. Line 176: “these invasive interactions” instead of “this invasive interactions”
1.17. Line 177: revise, e.g. “… in whole food webs to attempt …

Experimental design

The study’s research question are outlined at the end of the Introduction, and the performed systematic review and meta-analysis largely follow the standard approach, incl. being explicit about how the literature search was performed and how relevant papers were identified, a PRISMA statement and check list and a funnel plot. I have the following comment:
2.1. In Figure 1 (PRISMA flow diagram), something is not correct, as both of the two boxes on the right say that n=395 were excluded at that stage. This cannot be the case, so please indicate whether they were excluded in the screening stage or in the next stage when the full text was assessed.

Validity of the findings

I have the following comments on the validity of the findings:
3.1. Only eight relevant papers were identified with data about 17 invasive species, but 128 individual observations were extracted from these studies that were treated as independent data points in the analysis. It is a general problem with meta-analyses that if several observations are taken from a given study, these cannot be completely independent. I know that true independence does not really exist, but the authors should make sure that there are no strong dependencies between the 128 observations that they treat as being independent (keyword: pseudoreplication). This should be discussed and explicitly considered in the manuscript.
3.2. It might make sense to repeat the analysis at a species level, so that the data are pooled for each of the 17 invasive species. In this case, dependencies due to species identities can be excluded. I'd suggest to do that and compare the results to the current analysis, i.e. I'd keep the current analysis and additionally report results of the species-level analysis.

---

## Round 0.2 · accepted · Accept

Thank you for your detailed answers to the reviewers' comments. I consider you have successfully addressed all the concerns and appreciate the effort you made to improve the ms. Just note that you cited “Meza-Lopez & Siemann, 2015” in the text but not in the reference list. I’m happy to accept your ms and to move it forward into publication.